



# Soil degradation assessment across tropical grassland of Western Kenya

John N. Quinton[1*], Gabriel Yesuf[2], German Baldi[3], Mengyi Gong[4], Kelvin Kinuthia[5], Ellen L. Fry[6], Yuda Odongo[7], Barthelemew Nyakundi[7], Joseph Hitimana[7†], Patricia de Britto Costa[6], Alice A. Onyango[5], Sonja M. Leitner[5], Richard D. Bardgett[1,6], Mariana C. Rufino[8]

[1]Lancaster Environment Centre, Lancaster University, Lancaster, UK.
[2] Rural Payments Agency, Geospatial Services, Reading, UK
[3]Instituto de Matemática Aplicada San Luis –Universidad Nacional de San Luis & CONICET, San Luis, Argentina
[4]School of Mathematical Sciences, Lancaster University, Lancaster, UK
[5]Mazingira Centre for Environmental Research and Education, International Livestock Research Institute, Naivasha Rd, PO 30709, Nairobi, Kenya
[6]Department of Earth and Environmental Sciences, The University of Manchester, Oxford Road, Manchester, M13 9PT, UK.
[7]School of Agricultural Sciences and Natural Resources, University of Kabianga, P.O Box 2030 -20200.  Kericho, Kenya
[8]Chair of Livestock Systems, TUM School of Life Sciences, Building 4308, Liesel-Beckmann Straße 4, Freising 85354, Germany

†deceased
*Correspondence to: John N. Quinton (J.Quinton@lancaster.ac.uk)

## Abstract

Soils across sub-Saharan Africa are exposed to extensive degradation, reducing their ability to produce crops and support livestock. While there has been a significant research effort focussing on soil degradation in sub-Saharan croplands, less research effort had been directed towards grasslands. Here, we tested the effectiveness of remote sensing to classify the soil degradation status of smallholder grazing lands. Focussing on grasslands used by smallholders in the districts of Nyando and Kuresoi in Western Kenya, we first used remote sensing (RS) to classify grasslands as either equilibrium, transition or degraded, and then tested how this classification related to measured soil parameters indicative of soil degradation. We then used this classification, which was based on a temporal analysis of Normalised Differential Vegetation Index (NDVI), Enhanced Vegetation Index (EVI) and Normalised Differential Water Index (NDWI) between 2013 and 2018, to identify 90 field sites across the two districts, which we then sampled and analysed for a range of physical, chemical and biological soil properties. Only soil microbial biomass carbon (C) showed consistent alignment with the RS classification, although there was some overlap with other soil parameters at one or other of the sites. To group the sites using the soil parameters, which we split by district and into stable and transient soil variables, K-means clustering was undertaken. Two clusters were produced. One of the clusters included sites with higher levels of C, nitrogen (N), phosphorus (P) and pH, that aligned well with the RS classification at Kuresoi, with seven out of ten equilibrium sites being assigned to this cluster.  The other cluster, in Nyando, had high soil C and P, but low pH and relatively low soil bulk density,



and corresponded to 12 out of the 16 equilibrium sites. Overall, our results suggest that while the use of RS methods for classifying degraded grasslands and the soils supporting them does have significant advantages in terms of time and costs over field survey, supplementing these methods with a limited set of soil parameters related to nutrient cycling, such as microbial biomass C, soil P, percent C and N, and soil pH, could

enhance our ability to identify degraded soils and target restoration efforts.

**Introduction**

Approximately 660 million hectares of sub-Saharan African (SSA) soils are estimated to be degraded, which represents a significant portion of the global extent of degraded soils (Gibbs and Salmon, 2015). Soil degradation reduces the functioning of soils and is a

result of multiple processes including soil erosion by wind, water and tillage, salinisation, nutrient depletion, and compaction (Bridges and Oldeman, 1999) and may be triggered by shifts in land use, management or climatic changes. Most attention has been placed on the impacts of soil degradation on food security, and it has been cited as the leading cause of stagnation in food production, creating uncertainties for income

and nutritional security for rural populations (Barbier and Hochard, 2016). Reduced plant productivity associated with degraded soil also reduces the input of carbon (C) to the soil leading to lower C stocks (Bai and Cotrufo, 2022) and less biomass to support livestock. Further, when grazing lands are degraded, farmers are often forced to graze their livestock in adjacent forests, which can negatively affect forest plant communities

(Mullah et al., 2023). Thus, restoring degraded soils has become a priority for securing future food supply while simultaneously avoiding biodiversity and C losses. This has resulted in several initiatives supporting landscape restoration in Africa, notably the African forest restoration initiative (Messinger and Winterbottom, 2016), which gathered commitments from African governments to restore 100 million hectares of

degraded land by 2030.

The East African highlands of Kenya are densely populated areas of high agro-ecological potential. Farms here are small, typically smaller than 2 hectares (Lowder et al., 2016). Production includes a mix of grains and vegetables for local consumption, some cash crops, such as tea (*Camellia sinensis* (L.) O. Kuntze), and livestock keeping. Milk from

livestock is important to smallholder families as a valuable source of protein in a protein-poor diet (Hulett et al., 2014). Grazing animals are also culturally significant, reflecting the social standing of the owner and providing meat for celebrations and an additional source of cash when sold (Moll, 2005). Grazing animals include sheep, goats and cattle, and animal numbers range between 5-10 sheep and goats, and 1-2 cattle per

hectare. Additionally, grazing takes place on farms and on utility areas, which are controlled by local institutions; these often come under higher greater pressure because multiple livestock owners have access to the land. In response to these pressures, grassland soil degradation is widespread in Kenya (Nzau et al., 2018) although we know little about its extent and severity.

Given the importance of grazing land for sustaining rural livelihoods it is surprising that globally much less recent research attention has been placed on understanding degradation of grazing lands (Bardgett et al., 2021), particularly in SSA. High grazing pressures can degrade soil fertility with associated declines in soil properties underpinning soil health (Pelster et al., 2017), for instance causing soil compaction and

reducing soil infiltration rates (Owuor et al., 2018) and C inputs to soil due to the removal of plant material by livestock and reductions in root mass (Zhou et al., 2017).





Further, catchments with high livestock densities have larger nutrient and sediment loads in streams (Jacobs et al., 2017), have greater emissions of greenhouse gases (Arias-Navarro et al., 2017), and increase the risk of forest degradation (Brandt et al.,
2018). Low soil nutrient availability and the deterioration in soil physical properties impairs plant growth and alters plant nutrient concentrations (Augustine et al., 2003), and reduces organic matter return to soil. With poorer vegetation cover and lower organic matter contents, soils become increasingly vulnerable to erosion, leading to less soil depth and organic matter, which further reduces water and nutrient retention
(Quinton and Fiener., 2024). This leads to a downward spiral of productivity loss and reduced capacity of systems to resist and recover from climate extremes (Quinton and Fiener., 2024; Van De Koppel et al., 1997).

The UN Decade (2021-30) on ecosystem restoration (Unep, 2019) has focused attention on understanding where and how severely soils are degraded and whether they can
recover, which is clearly important for the design of restoration programmes. In grazed systems, soil degradation is often recognised by the presence of bare soil. However, using bare soil as an indicator can be problematic in systems where erratic rainfall patterns lead to seasonal and inter-annual fluctuations in vegetation growth coupled with reduced vegetation cover due to grazing (Ellis and Swift, 1988). In such
environments, poor vegetation growth may or may not indicate degraded soils. However, utilising the response of vegetation to changed soil properties and water availability is an approach that has been used by several authors (e.g. Eckert et al., 2015; Zhou et al., 2017).

Here, we tested the reliability of remote sensing approaches for classifying degradation
status of smallholder grazing land and compared it with an approach based on the sampling of soils and characterisation of soil properties related to soil structural stability and C, nitrogen (N) and phosphorus (P) cycling. Working in two areas representing smallholder grazing land of western Kenya (Nyando and Kuresoi), we assessed degradation using a dynamic multi-year approach to derive a range of metrics
to quantify the magnitude, seasonality and interannual variability of the vegetation (Rufino et al., 2016), and then tested whether or not the classification was related to measured soil parameters. We then explored whether soil variables classified as either stable or transient could be used to classify soil degradation status in grasslands.

**Methodology**

**Field areas**

We used a comparative landscape-level analysis of two agro-ecosystems with different ecologies (Figure 1). The sites are in western Kenya covering the neighbouring basins of the rivers Sondu-Miriu and Nyando spanning land use transitions from East African montane forests to grasslands and croplands. Site 1 (Kuresoi) is in Kericho county
located in the Sondu river basin in the proximity of the Mau Forest, at an altitude ranging from 1,700 to 3,000 masl, with an average rainfall of 1,988±328 mm. The geology originates from the early Miocene, with phonolites dominating in the lower part of the catchment, and phonolitic nephelinites in the upper part. A variety of Tertiary tuffs are found on the highest part of the Mau Escarpment (Jennings, 1971). Site 2 is in
Lower Nyando located in the Nyando river basin, with an average rainfall of 1,150 mm and spanning from the foot of a plateau at 1,600 towards Lake Victoria at 1,200 masl. Soils are derived from Holocene alluvial deposits, and a variety of parent materials including phonolites and granitic gneisses (Iuss, 2015). The Lower Nyando site covers





an area which is approximately 160 km², whilst the Kuresoi site covers an area that is
approximately 1,300 km² next to the Mau Forest. More details on land use and
vegetation are given below.

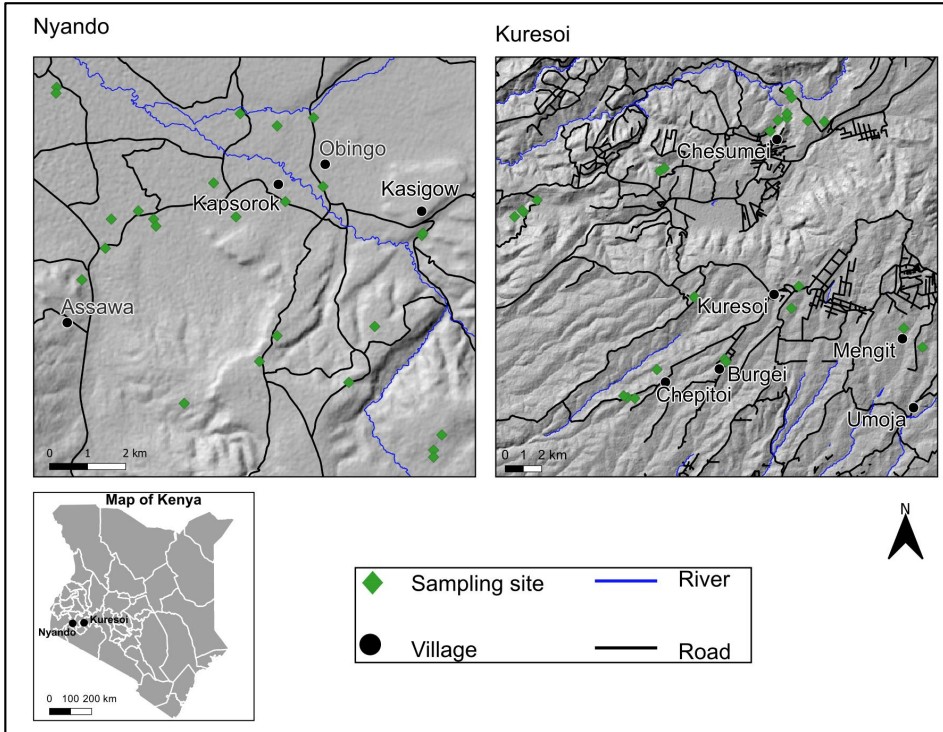

**Figure 1:** Location map showing project sites in Kenya (bottom left) and expanded
views of Kuresoi (top right) and Lower Nyando (top left). Map produced using ArcGIS®
software by Esri. Road network, river and settlement were reproduced using
OpenStreetMap vector data. Accessed on 2019-06-09 and are licensed under the Open
Database 1.0 License. Digital Elevation Model was produced using ASTER Global Digital
Elevation Model (GDEM) 30-meter resolution as input and under license from NASA
Earth Science Information Partners Data Preservation and Stewardship Committee.
2019. Earth Science Data. Ver. 2.

The first step to define degradation states following the concepts developed by (Briske
et al., 2003) involved an analytical approach using remote sensing images of both study
sites. This spatio-temporal analysis covered a period of 5 years (2013 – 2018), where
productive grazing lands were classified as being in equilibrium, grazing lands that
followed a variable trend were defined as transition, and degraded grazing land were
those shown as unstable and unproductive.

**Remote sensing data selection**
To classify degradation status, 35 satellite image scenes were collected from the archives
of European Space Agency (Esa, 2016) and the United States Geological Surveys earth
explorer (Usgs, 2025) (Supplementary Table S1). The selection of different sensors was
necessary to fill missing dates from the Sentinel collection which had a higher spatial





resolution (10 m) but has been deployed in space for a shorter period (since 2015) compared to the Landsat sensors. The final satellite imagery was from Landsat-Thematic Mapper (TM), Landsat Operational Land Imager (OLI) and Sentinel-2 sensors. Landsat-TM and OLI imagery have a spatial resolution of 30 m. The mapping of grasslands in smallholder production systems required high-resolution imagery because of the relatively small sizes of the fields, which are often less than 1 ha. Therefore, the Landsat-derived images were resampled to 10 m using the Sentinel-2 imagery as reference.

**Temporal and seasonal analysis**

Three vegetation indices, Normalised Differential Vegetation Index (NDVI), Enhanced Vegetation Index (EVI) and Normalised Differential Water Index (NDWI) were calculated using blue, red, near infra-red (NIR), and shortwave infra-red bands (Equations, 1, 2 and 3). These indices were selected because vegetation and water indices are effective to estimate changes in ecosystems (He et al., 2018) and grassland biomass (Todd et al., 1998), distinguish canopy density (Huete et al., 1997), and characterise drought (Rulinda et al., 2012).

$$NDVI = \frac{(NIR - Red)}{(NIR + Red)} \qquad (1)$$

$$EVI = G * \left[ \frac{(NIR - Red)}{(NIR + C1 * Red - C2 * Blue + L)} \right] \qquad (2)$$

$$NDWI = \frac{(NIR - SWIR)}{(NIR + SWIR)} \qquad (3)$$

where $NIR$ is near-infra red; $G$ represents a gain factor; $L$ adjusts for canopy background; $C_1$ and $C_2$ are coefficients for atmospheric resistance ($G$ = 2.5, $C_1$ = 6, and $C_2$ = 7.5). Applying these coefficients allows for index calculation as a ratio between Red and $NIR$ values, while reducing the background noise, atmospheric noise, and data saturation. Index values were calculated on a scale of -1 to 1.

The seasonality of the vegetation was interpolated using TIMESAT v3.3 algorithm (Eklundh and Jönsson, 2015). An adaptive Savitzky-Golay smoothed function was fitted over the time-series of Lower Nyando to model bi-modal seasons and to determine the timings of the growing seasons. A double gaussian function was fitted over the time-series of Kuresoi to model seasonal peaks where the vegetation dynamics is less variable. The adaptive function of TIMESAT modelled abrupt changes in vegetation effectively, which was often the case in the Lower Nyando landscape consisting of an intricate mosaic of land covers. A double logistic function allowed to isolate noise (e.g. caused by clouds) in Kuresoi data. To capture seasonal peaks, the functions were fitted to the upper envelope of the time-series following Eklundh and Jönsson (2015). After fitting the statistical functions to the data, the following seasonal parameters were estimated: Seasonal Amplitude (Amp), Start of Season (StoSt), End of Season (EoS), Function value at Start of Season (SoSv), Function value at End of Season (EoSv), Season Length (Len), Base level, Mid of the Season (Mid), Largest data, Maximum Value (MV), Left Derivative/greening rates (LD/GR, and Right Derivative/browning rates (RD/BR), Large Seasonal Integral and Small Seasonal Integral. For definitions of seasonal parameters and further explanations see (Eklundh and Jönsson, 2017).

**Degradation units' classification**

Six seasonal parameters were selected for the classification of degradation units: SoSv, EoSv, MV, GR and BR because of the phenology characteristic of the ecosystems under study (Kong et al., 2022). Vegetation at equilibrium state was expected to have higher values for SoSv, EoSv, MV and experience faster greening compared to vegetation of the





units with transition and regime-shift states (Xiao et al., 2006; Yu et al., 2012). There are no predefined seasonal parameter values that define the stages of grassland degradation in Western Kenya. For instance, Tagesson et al. (2015) quantified maximum NDVI values between 0.59 and 0.82 for different grasslands in a semi-arid region of Senegal. Therefore, thresholds were defined using the average distribution of the selected

seasonal parameters (Table 1). Thresholding was implemented using written functions in R to partition parameter values into three groups corresponding to equilibrium, transition, and regime-shifts. Several models were generated using different combinations of seasonal parameter classifications (Table 1). However, only two models were visually consistent with the spatial distribution of dominant land cover types (e.g.,

large forest patch). All land cover types were retained during seasonal parameter estimation and classification to allow for accurate seasonal models of the sites. Using the above approach and thresholds, vegetation at equilibrium state was assigned to high MV(>0.8), high GR (>0.5), and low BR (<0.3). Vegetation at the regime-shift state had low MV (<0.5), low GR (≤0.8), and high BR (≤ 0.5). Finally, the classification from each

index was combined to determine areas of common agreement.

**Table 1:** Summary of models' description, seasonal parameters and threshold values used for degradation unit classification of Lower Nyando and Kuresoi.

| Description | Index | Threshold values (Nyando)§ | | | | | Threshold values (Kuresoi)§ | | | | |
|---|---|---|---|---|---|---|---|---|---|---|---|
| | | MV | SoSv | EoSv | GR | BR | MV | SoSv | EoSv | GR | BR |
| Model 1 | NDVI | 0.54 | 0.55 | 0.55 | 0.58 | 0.56 | 0.81 | 0.78 | 0.81 | 0.8 | 0.78 |
| | EVI | 0.49 | 0.5 | 0.49 | 0.54 | 0.54 | 0.75 | 0.76 | 0.76 | 0.79 | 0.74 |
| | NDWI | 0.81 | 0.79 | 0.79 | 0.77 | 0.81 | 0.83 | 0.82 | 0.82 | 0.79 | 0.80 |
| Model 2 | NDVI | 0.54 | 0.55 | 0.55 | 0.58 | | 0.81 | 0.78 | 0.81 | 0.80 | |
| | EVI | 0.49 | 0.50 | 0.49 | 0.54 | | 0.75 | 0.76 | 0.76 | 0.79 | |
| Model 3* | NDVI | 0.54 | 0.55 | 0.55 | | | 0.81 | 0.78 | 0.81 | | |
| | NDWI | 0.81 | 0.79 | 0.79 | | | 0.83 | 0.82 | 0.82 | | |
| Model 4 | EVI | 0.49 | 0.50 | 0.49 | 0.54 | 0.54 | 0.75 | 0.76 | 0.76 | 0.79 | 0.74 |
| | NDWI | 0.81 | 0.79 | 0.79 | 0.77 | 0.81 | 0.83 | 0.82 | 0.82 | 0.79 | 0.80 |
| Model 5** | NDWI | 0.81 | 0.79 | 0.79 | | | 0.83 | 0.82 | 0.82 | | |

* More consistent classification of degraded grasslands and bare grounds as regime-shift in Nyando
** More consistent classification of grasslands at equilibrium in Kuresoi
§ Threshold values represent the average distributions.

**Selecting sampling locations**

Through visual inspection, model 3 and model 5 were found to be the most consistent with Google Earth Images of the sites. Subsequently, the land cover data from the European Space Agency (ESA 2016) was used to mask forests, urban and water bodies to detect grazing areas. Afterwards, locations were selected using the Fishnet tool of ArcGIS. Stratified random sampling was used to create sampling locations separated by

a minimum distance of 1 km to select approximately 30 sampling locations for each degradation unit, resulting in 100 sampling locations including replacements. The status of the locations was checked visually in Google Earth (2008 - 2018) to remove locations that coincided with recently cultivated areas (<10 years) and/or road tracks. Locations



that had signs of recent cultivation or tillage lines were excluded. In October-November 2019, the locations were visited to remove sample locations that were inaccessible or when landowners denied access. In total, 90 sites were finally selected after land use history checks and obtaining consent of farmers/landowners (Figure 2).

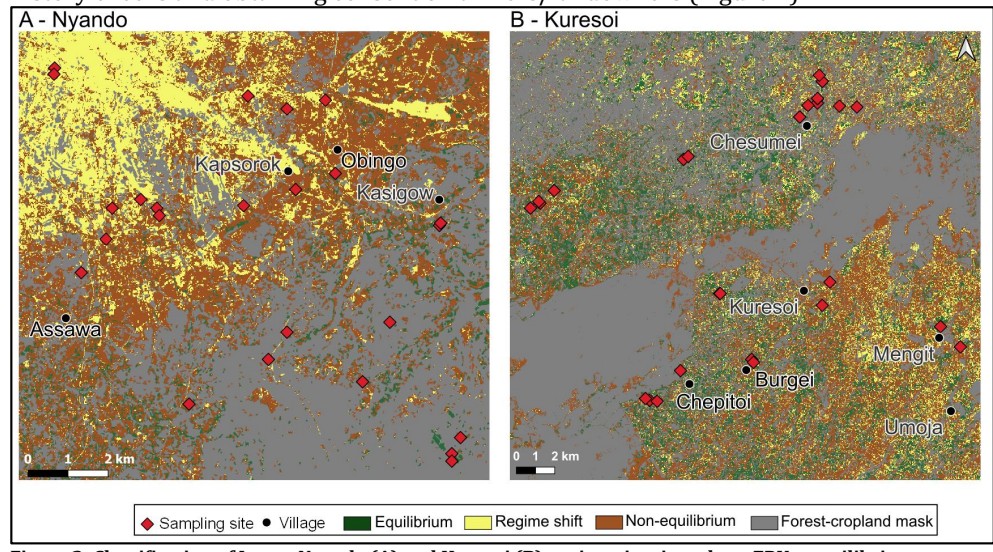

**Figure 2: Classification of Lower Nyando (A) and Kuresoi (B) project sites into three ERUs: equilibrium, transition and regime-shift. Sampling sites are overlaid and show the distribution of field experiments and locations of soil and aboveground biomass samples. Map produced using ArcGIS® software by Esri. Road settlement information were reproduced using OpenStreetMap vector data. Accessed on 2019-06-09 and are licensed under the Open Database 1.0 License. .**

**Soil sampling and analyses**

Soils at each site were sampled to 10 cm depth and analysed for a range of physical, chemical and biological parameters (Table S1).

Bulk density was calculated following sampling of intact soil with 45 mm diameter rings and soil texture was determined by laser diffraction (Beckman-Coulter LSI3 320), after soil dispersion in sodium hexametaphosphate. Aggregate stability was determined using

the fast-wetting method of aggregate stability (Le Bissonnais, 1996), which subjects the aggregates to rapid immersion in water for 10 min. After that, aggregate samples were sieved in ethanol before oven drying to determine final aggregate size distribution, producing a mean weight diameter (MWD).

For each sampled plot we measured total soil C, N, P, and selected microbial-mediated

functions related to nutrient cycling  These were microbial biomass (C and N), nutrient availability (i.e. soluble inorganic and organic N and P pools, and dissolved organic C), rates of N mineralisation and nitrification, and a suite of extracellular enzyme activities involved in the degradation of cellulose, chitin, lignin and proteins (i.e. β-glucosidase (GLC), cellobiohydrolase (CBH), β-xylosidase (XYL), N-acetylglucosaminidase (NAG),

phosphatase (PHO), phenol oxidase (POX), peroxidase (PER), and urease (URE)), following Fry et al. (2018) and De Vries and Bardgett (2016), and as described in Broadbent et al. (2022) for extracellular enzymes.

Briefly, percentage C and N in dry, ground soil were measured using an elemental combustion analyser (Elementar Vario EL, Hanau, Germany). We measured dissolved

total and organic C (DC and DOC respectively), plant available nitrate ($NO_3^-$) and total dissolved N (TDN) by weighing 5g of fresh soil accurately and shaking in 35ml Milli-Q





water for 10 minutes at 150rpm, before filtering through Whatman 42 filter paper. C in the filtrate was quantified using an Aurora 1030W TOC analyser (OI Analytical, UK), and N was quantified using an autoanalyser (AA3, Seal Analytical, Wrexham UK). Organic N
was calculated by subtracting inorganic N values (nitrate and ammonium) from total N. pH of the filtrate was determined using a pH probe (Mettler Toledo FE20, Salford, UK). Values were adjusted for soil moisture. Soil ammonium ($NH_4^+$) was measured by shaking 5g of fresh soil in 25ml 1M KCl for 30 minutes, extracting through Whatman 1 filter paper and analysing on the autoanalyser as before. For potential mineralisation
and nitrification, 5g of each soil sample was incubated for 14 days at 25 °C before being extracted and analysed for $NH_4^+$ and $NO_3^-$ using the KCl procedure. The values from the initial KCl extraction (summed $NH_4^+$ and $NO_3^-$) were subtracted from the day 14 extraction and divided by 14 to give a rate of potential mineralisation per day. Nitrification was calculated by using the $NO_3^-$ values only. Negative values imply
denitrification, i.e. loss of N as $N_2$ gas. Microbial biomass C and N were determined using the chloroform-fumigation method (Vance 1987). We weighed 5 g of each sample twice. The first replicates were shaken in 25 ml 0.5M $K_2SO_4$ for 30 minutes, before passing through Whatman 42 filter paper. The second were placed in a desiccator containing a beaker of chloroform under vacuum for 24 hours to lyse microbial cells, before being
extracted as before. Total dissolved C and total extractable N were analysed using the Aurora and the autoanalyser respectively. Microbial biomass C and N were calculated by subtracting the unfumigated values from the fumigated ones.
Total soil P was measured using the Kjeldahl digestion method (Kjeldahl, 1883). We mixed 420 ml concentrated sulfuric acid with 12 g lithium sulphate. We added 0.5 ml of
this mixture to 50 mg of dry ground soil per sample in glass digestion tubes. We then added 0.5 ml 30% hydrogen peroxide. Samples were heated at 200°C, then we added a 50°C heat increase every 30 minutes until it reached 360°C. Samples were heated at 360°C for two hours before cooling. When cool, 0.5ml of hydrogen peroxide was added and samples were digested at 360°C for a further two hours. Samples were diluted to
50ml using Milli-Q water. They were analysed using the ascorbic acid microplate method after (Kuo, 1996), where samples were measured colourimetrically at 880 nm. For inorganic P, we placed 2g of dry soil into a falcon tube with 50ml of 0.5M sulfuric acid. This was shaken at 150rpm for 16 hours. The samples were centrifuged at 1500 rpm for 10 minutes, and the supernatant was analysed using the ascorbic acid method (Olsen
and Sommers, 1982). Broadbent et al. (2022)

**Description of the data set**
For testing and clustering analysis, we focused on a total of 28 variables measured from the samples collected from the 0-0.1m depth in Kuresoi and Nyando, respectively. These variables were grouped as either stable or transient soil variables and relate to the rate
of change of these parameters in response to degradation. Changes in bulk density and soil hydraulic properties can persist over seasonal to multi-annual timescales (Berisso et al., 2012), as can contents of C, N and P along with pH, aggregate stability, sand percentage, silt percentage, and clay percentage; hence, these parameters were considered to be stable soil variables. In contrast, soil biological parameters, including
enzyme activities, microbial biomass, and rates of nutrient mineralisation, respond rapidly to change in environmental conditions (Cordero et al., 2023) and therefore soil enzymes ( PHO, GLC, NAG, XYL, CBH, PER, POX, URE), water extractable $NO_3$, and KCl-extracted $NH_4$, microbial C, microbial N, total dissolved C, organic dissolved C, mineralisation and nitrification were considered transient. Finally, the sites with



incomplete data (i.e., with missing observation in any of the variables in the stable or
      transient variable sets) were removed, resulting in 31 sites in Nyando, 41 sites in
      Kuresoi for the stable variables, and 42 sites in Nyando, 38 sites in Kuresoi for the
      transient variables. The distribution of degradation states of these sites is summarised
      in Table S2.

**Statistical analysis of field data**
      Statistical analyses were carried out to investigate differences in field sampling data
      between sites with different degradation labels allocated from remote sensing (Table 2).
      First, analysis of variance (ANOVA) was applied to all soil variables to identify any mean
      differences between the degradation classes within Kuresoi and Nyando respectively.
For soil variables with a significant mean difference, t-test was then applied to each pair
      of degradation classes (e.g., equilibrium vs.. degraded, transition vs.. degraded) to
      further investigate the content of the mean differences.

      **Description of the clustering methods**
      Considering the features of our data sets, i.e., relatively large number of variables as
compared to the number of sites and relatively high variability in some variables, and
      the initial experiments with different clustering methods, we chose to use k-means
      clustering for our main analysis. In particular, the k-means clustering was applied to the
      principal components extracted from the data. We also applied the Gaussian mixture
      model to the data sets, but only for reference. Below we briefly introduce the clustering
methods and provide some details on the approach we took.
      K-means clustering is a popular method for grouping a population of n subjects (n being
      the number of sites in this case), each of p-dimensional (p being the number of
      covariates), into a number of k clusters, using algorithms developed by e.g.,(Hartigan
      and Wong, 1979; Lloyd, 1957; Macqueen, 1967). Gaussian mixture model is another
popular clustering approach. It is a model-based clustering method introduced by Fraley
      and Raftery (2002), where it assumes that the population follows a mixture of k p-
      dimensional Gaussian distributions. Few assumptions are required for applying the k-
      means algorithm, although it has been acknowledged that the method works better with
      clusters that are of similar shapes or sizes (Steinley, 2006). The result can be sensitive to
outliers (Johnson and Wichern, 2007). In contrast, a Gaussian mixture model needs
      specific assumptions on the covariance structure, some of which involve the estimation
      of a large number of parameters and hence is not suitable for a small sample size.
      Considering these features, k-means clustering seems to be a more suitable choice over
      Gaussian mixture model when it comes to data sets with high dimensionality, high
variability and relatively small sample size.
      To determine the number of clusters for k-means clustering, methods such as elbow plot
      of the total sum of squared distance between points and cluster centres and gap
      statistics (Tibshirani et al., 2001) can be used. For Gaussian mixture model, Bayesian
      information criterion (BIC) can be used to select cluster numbers  (Scrucca et al., 2023),
providing a more objective solution. Due to the relatively small population size in this
      analysis, only cluster numbers from two to five were investigated. Based on the model
      selection criteria, after taking the robustness of the clustering results into account and
      discounting the cluster numbers  thatced singletons (i.e., one site as a group of its own),
      the cluster number was settled to be two for both stable and transient variable sets in
Kuresoi and Nyando, respectively.



Due to the relatively large number of variables (p) as compared to the relatively small number of sites (n), principal component analysis (PCA) was first applied to reduce the dimension of the data. The number of principal components (PCs) was selected to account for at least 80% of the information in the data, or the correlation matrix to be
precise. This resulted in a much smaller number (six) of "variables" in the form of PCs to be used in the clustering analysis, which helped to improve the stability of the clustering algorithms.

As the analysis was purely exploratory, it was carried out as if we had little prior knowledge on the subject (i.e., we did not use the degradation states to guide the
clustering analysis). We explored the results from three different clustering methods and used Rand index (Rand, 1971) to investigate the correspondence between the results. In this case, the Rand indices were moderate in the clustering of stable and transient variables. This suggested that the two clustering methods agreed to some extent. Finally, t-tests were applied to see if one clustering result separated the
population better, and the results from the k-means clustering appeared to perform better in this case. Considering all the analyses and tests above, we chose to focus on discussing the clustering results from the k-means in the next section.

The clustering analysis was implemented in R using the "kmeans" function from base R (R. Core Team, 2023) and the "mclustBIC", "Mclust" functions from the "mclust" package
(Scrucca et al., 2023). Comparison of the clustering results was carried out using the "rand.index" function from the "fossil" package (Vavrek, 2011).

**Results**

**Relation of remote sensing classification to measured soil parameters**

Microbial biomass C and soil bulk density were the only two variables that showed
significant differences (p<0.1) between degradation classes at both areas. There was a significant increase (p<0.05) of 74% in mean microbial biomass C from degraded sites to equilibrium sites at Kuresoi, and a significant increase (p<0.05) of 70% at Nyando. Although the differences between the transition and degraded/equilibrium sites were not significant, the rankings of the class means were consistent
(degraded<transition<equilibrium) for both areas. The largest difference in soil bulk density was seen between the transition class and the equilibrium class for both Kuresoi (p<0.05) and Nyando (p<0.05). In this case, the rankings are inconsistent, with equilibrium>degraded>transition at Kuresoi and transition>degraded>equilibrium at Nyando, although the absolute differences between the classes were small (c.f. 0.1 g
cm3). Of the other soil variables that showed significant differences between degradation classes within each area, only pH, total N and C and XYL at Kuresoi and C:N ratio at Nyando ranked the classes in the order degraded<transition<equilibrium. Specifically: at Kuresoi, mean pH increased by 0.4 from degrade class to equilibrium class, mean total C increased from 6.1% to 7.9%, mean total N from 0.5% to 0.7%, and
mean XYL increased by approximately 54%, from 172.9 to. 267.6 nmol h-1 g-1 dry soil; at Nyando C:N ratio increased from 12.1 in the degraded class to 13.2 in the equilibrium class.





**Table 2** Table of mean and standard deviation (in brackets) of 28 soil variables (top 0-10 cm) by degradation labels measured at field sites in two locations, Kuresoi and Nyando, along with the significance level of the between label differences from ANOVA (where * represents $p < 0.1$, and ** represents $p < 0.05$). Variables that show a significant difference between degradation classes at the p<0.1 level as determined by a pairwise t-test are designated by a different letter in parenthesise.

| Variable | Kuresoi | | | ANOVA | Nyando | | | ANOVA |
|---|---|---|---|---|---|---|---|---|
| | Degraded | Transition | Equilibrium | | Degraded | Transition | Equilibrium | |
| pH | 5.0 ±0.5 (a) | 5.3±0.5 (b) | 5.4±0.5 (b) | * | 5.8±0.9 | 5.4±0.6 | 5.6±0.8 | |
| Total inorganic N (mg kg$^{-1}$) | 21.7±14.8 | 21.0±13.4 | 31.7±18.9 | | 9.9±7.0 (a) | 23.7±16.3 (b) | 13.6±6.7 (a) | ** |
| Organic N (mg kg$^{-1}$) | 7.2±4.1 | 8.0±3.3 | 8.1±3.3 | | 6.1±2.9 (a) | 9.3±3.6 (b) | 8.9±3.7 (b) | * |
| Inorganic P (mg kg$^{-1}$) | 131.4±147.5 | 82.7±67.7 | 70.9±82.5 | | 128.8±86.4 | 124.1±60.2 | 90.4±68.7 | |
| Total P (mg kg$^{-1}$) | 1093.4±568.3 | 1176.1±631.2 | 1328.2±729.3 | | 660.2±421.5 | 457.0±188.0 | 523.3±367.9 | |
| Total N (%) | 0.5±0.1 (a) | 0.6±0.2 (ab) | 0.7±0.2 (b) | * | 0.2±0.1 | 0.3±0.1 | 0.3±0.1 | |
| Total C (%) | 6.1±1.7 (a) | 6.8±2.3 (ab) | 7.9±2.3 (b) | * | 2.8±1.3 | 4.1±1.4 | 3.8±1.9 | |
| Soil bulk density (g cm-3) | 0.8±0.1 (a) | 0.7±0.1 (ab) | 0.8±0.1 (b) | ** | 1.0±0.0 (a) | 1.1±0.1 (b) | 0.9±0.1 (c) | ** |
| Aggregate stability Mean weight diameter (μm) | 314.1±54.3 | 321.3±62.6 | 312.0±40.9 | | 279.2±56.7 | 234.2±111.9 | 245.5±119.5 | |
| Sand (%) | 7.8±9.6 | 9.3±11.0 | 11.7±8.6 | | 18.6±18.5 | 22.7±20.7 | 15.4±15.4 | |



| | | | | | | | |
|---|---|---|---|---|---|---|---|
| Silt (%) | 60.7±12.4 | 63.1±9.9 | 63.0±7.0 | 57.3±15.0 (ab) | 49.8±19.0 (a) | 65.0±12.5 (b) | ** |
| Clay (%) | 31.4±14.5 | 27.6±11.1 | 25.3±8.8 | 24.1±13.8 | 27.5±20.8 | 19.5±8.1 | |
| PHO (nmol h⁻¹ g⁻¹ dry soil) | 3094.3±1130.5 | 3708.3±1236.4 | 3797.6±1408.5 | 1966.7±878.1 | 2701.1±1372.4 | 2160.2±1419.5 | |
| GLC (nmol h⁻¹ g⁻¹ dry soil) | 200.0±131.4 | 221.2±117.4 | 269.4±122.4 | 229.3±104.7 | 349±226.625 | 230.6±231.6 | |
| N NAG (nmol h⁻¹ g⁻¹ dry soil) | 82.2±41.0 | 99.3±56.5 | 83.4±44.4 | 78.4±45.5 | 97.7±40.3 | 72.5±30.2 | |
| XYL (nmol h⁻¹ g⁻¹ dry soil) | 172.9±92.6 (a) | 196.0±86.2 (ab) | 267.6±157.1 (b) | 197.8±86.9 | 259.2±166.5 | 175.2±148.9 | * |
| CBH (nmol h⁻¹ g⁻¹ dry soil) | 34.3±11.0 | 42.7±14.6 | 41.9±23.6 | 33.5±27.2 | 36.4±26.0 | 44.4±47.2 | |
| PER (nmol h⁻¹ g⁻¹ dry soil) | 11.1±14.4 | 8.9±8.5 | 6.9±7.4 | 4.1±3.0 | 3.2±1.9 | 4.8±2.6 | |
| POX (nmol h⁻¹ g⁻¹ dry soil) | 0.3±0.2 | 0.3±0.3 | 0.2±0.2 | 0.3±0.1 (a) | 0.2±0.1 (b) | 0.3±0.1 (a) | ** |
| URE (nmol h⁻¹ g⁻¹ dry soil) | 8.7±5.7 | 6.9±6.1 | 10.5±6.8 | 8.0±7.5 | 12.3±8.9 | 7.3±5.8 | |
| KClNH₄ (mg kg⁻¹) | 9.5±17.5 | 8.1±7.4 | 9.3±13.7 | 7.0±4.6 | 9.4±9.3 | 8.5±7.0 | |
| H₂ONO₃ (mg kg⁻¹) | 13.3±14.4 | 12.3±11.5 | 22.4±18.8 | 3.9±4.9 | 13.6±15.9 | 7.8±17.8 | |





| Variable | | | | | | | |
|---|---|---|---|---|---|---|---|
| Microbial C (mg kg$^{-1}$) | 1486.7±1171.9 (a) | 1760.0±1310.0 (ab) | 2583.0±1095.6 (b) | * | 863.6±568.0 (a) | 1167.7±618.6 (ab) | 1471.4±676.2 (b) ** |
| Microbial N (mg kg$^{-1}$) | 109.9±53.7 | 126.8±92.5 | 171.2±107.1 | | 67.1±40.2 | 95.3±56.9 | 84.0±50.0 |
| Dissolved Total C (mg kg$^{-1}$) | 267.7±73.2 | 296.3±85.7 | 288.1±96.5 | | 293.3±95.9 | 319.3±82.4 | 338.8±152.2 |
| Dissolved Organic C (mg kg$^{-1}$) | 263.7±73.3 | 293.3±86.8 | 283.4±93.5 | | 269.0±83.3 | 301.5±71.7 | 322.6±145.1 |
| Mineralisation (mg kg$^{-1}$ d$^{-1}$) | 0.2±1.1 | -0.0±0.8 | -0.5±1.6 | | -0.2±1.0 | -0.2±3.3 | 0.6±0.6 |
| Nitrification (mg kg$^{-1}$ d$^{-1}$) | -0.1±1.1 | -0.7±0.9 | -1.2±1.7 | | -0.2±1.1 | -0.1±2.7 | -0.1±0.7 |
| CN ratio | 12.0 ± 1.1 | 12.1 ± 0.8 | 11.5 ± 0.7 | | 12.1 ± 1.2 (a) | 12.4 ± 1.3 (a) | 13.2 ± 0.8 (b) ** |
| CP ratio | 76.8 ± 56.3 | 74.6 ± 45.3 | 120.1 ± 166.4 | | 83.9 ± 105.2 | 125.6 ± 109.7 | 98.5 ± 62.1 |
| NP ratio | 6.5 ± 5.0 | 6.1 ± 3.7 | 10.4 ± 14.3 | | 7.3 ± 9.8 | 10.5 ± 9.8 | 7.6 ± 4.9 |





**Which stable and transient soil variables explain the clustering of soils in the two study sites?**

Table 3 summarises the number of sites in Kuresoi and Nyando that have been grouped
into two clusters by the k-means algorithm for stable and transient variables. Note that the total number of sites used in each clustering analysis is different.

**Table 3: Number of Stable and transient sites allotted to cluster 1 and two at Kuresoi and Nyando using K-means clustering. .**

|  | Kuresoi | | Nyando | |
|---|---|---|---|---|
|  | Cluster 1 | Cluster 2 | Cluster 1 | Cluster 2 |
| Stable | 18 | 23 | 10 | 21 |
| Transient | 16 | 22 | 15 | 27 |

For the stable variables, in Kuresoi, sites in cluster 2 had significantly higher values of total N, total inorganic N, organic N, total P, total C and pH. There was a significant difference in silt and clay contents of the two clusters. We found that 7 out of 9 Kuresoi equilibrium sites, from the remote sensing classification, were assigned to this cluster 2, but the numbers of transitional and degraded sites were distributed evenly between two

clusters. Similarly, in Nyando, one cluster (Cluster 2) tended to have higher levels of total P, total N and total C, but lower pH and relatively low soil bulk density. There was a significant difference in sand, silt and clay percentages. In total, 12 out of 16 equilibrium sites in Nyando were assigned to this cluster. The transitional and degraded sites appeared to be equally likely in two clusters. Density plots showing how the two

clusters differ in selected stable variables are given in the top five panels in Figures 3 (Kuresoi) and 4 (Nyando).

For the transient variables, in Kuresoi, sites in one cluster (Cluster 1) tended to have higher PHO, GLC, XYL, CBH, but lower POX. It also had higher microbial N, nitrate (extracted in $H_2O$ $NO_3$), microbial C, total dissolved C. In Nyando, one cluster (Cluster 1)

consisted of sites with higher PHO, GLC, XYL, NAG, total dissolved C, but lower PER and POX. The cluster labels did not match the degradation labels in both cases. This is not surprising as the transient variables are highly variable and can change substantially in a short period of time. Density plots showing how the two clusters differed in selected transient variables are given in the bottom panels of Figures 3 (Kuresoi) and 4

(Nyando).





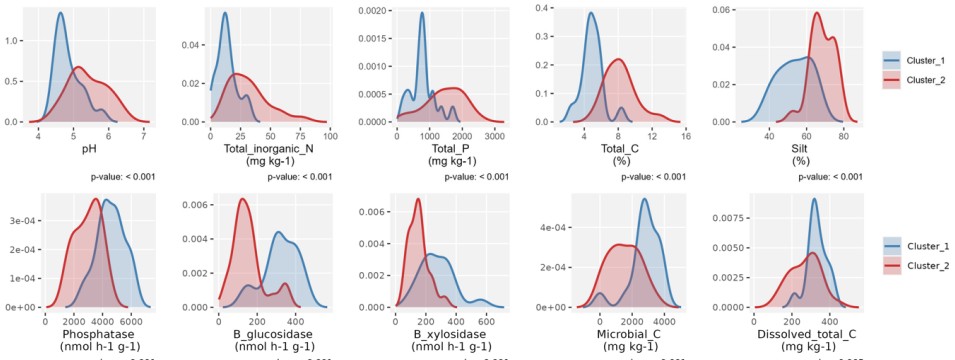

**Figure 3. Density plots of selected stable variables from sites that are grouped into two clusters (top panels) and transient variables from sites that are grouped into two clusters (bottom panels) in Kuresoi.**

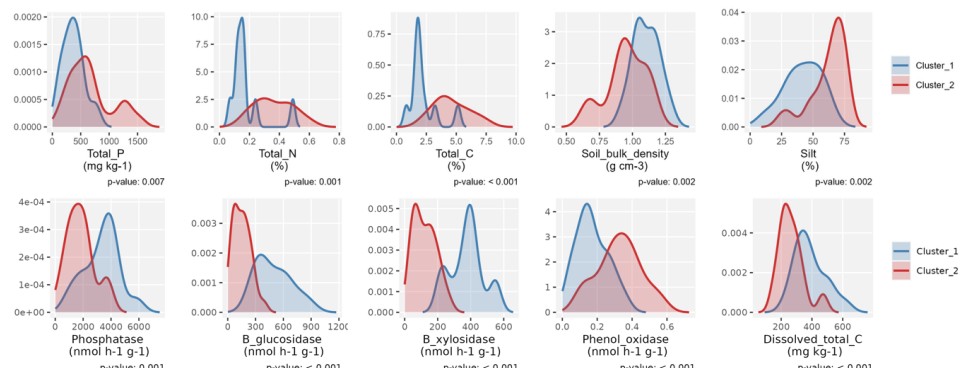

**Figure 4. Density plots of selected stable variables from sites that are grouped into two clusters (top panels) and transient variables from sites that are grouped into two clusters (bottom panels) in Nyando.**

## Discussion

Remote sensing is a powerful tool to assess soil degradation and has been utilised globally in many studies (e.g. Cordell et al., 2017; Manić et al., 2022; Wang et al., 2024) The ability to classify degradation status over large areas, at relatively low cost and utilising data that can be rapidly updated as new images become available, is an attractive proposition, since it provides land managers, policy makers and scientists with a mechanism for targeting interventions. Soil mapping of soil properties and soil degradation has combined remote sensing and measured soil properties to map soils in Africa with some success (Vågen et al., 2016). Nevertheless, there have been relatively few attempts to compare remotely sensed classification against soil data collected from soil sampling programmes. Our work demonstrates that, while it is relatively straightforward to generate classifications using derived parameters, such as NDVI, NDWI and EVI, that reflect vegetation dynamics, the resulting classification poorly reflects changes in multiple in-situ soil parameters related to soil degradation. However, across the two studied districts (i.e., Nyando and Kuresoi) we detected consistent alignment between remote sensing classification of degradation and microbial biomass C, a key soil biological parameter related to nutrient and C cycling processes in soil



(Tate, 2017) that is tightly linked to plant diversity and productivity (Chen et al., 2019)
and is known to respond quickly (c.100 days) to inputs of fresh organic matter to soil,
including plant litter and animal wastes (Dai et al., 2021).  Therefore, it is likely that we
are seeing the soil response to the amount of litter, root exudates, and dung from grazing
animals, that is returned to the soil, all of which are functions of above-ground biomass
reflected in the dynamics of NDVI.
Apart from microbial biomass C, there was little consistent agreement between the
remotely sensed classification with field-based soil variables (Table 2).  Some variables
related to soil degradation, such as C and N concentrations, C:N ratio and pH, were
statistically significant for one site, but not the other. Soil C and N concentrations are
considered good proxy indicators for soil health and are correlated with other
important soil functions (Lal, 2016), and they were lower in degraded soils in Burkina
Faso compared to those under native vegetation (Traoré et al., 2015). However, in our
study all sampled sites are managed grasslands, providing less of a contrast.
An additional problem with the RS classification is the difficulty associated with
unravelling the effect of rainfall variability and soil degradation (Wessels et al., 2007).
These difficulties are compounded in the context of smallholder farming due to grazing
occurring on small parcels of land where plant biomass is variable and depends not only
on soil and rainfall, but upon frequency and intensity of grazing. Thus, in these
situations counter-intuitive results are possible. For example, following a drought it is
likely that grazing takes place on the most resilient and rapidly recovering areas, the
equilibrium and transitional plots in this study, rather than those that are slow to
revegetate, potentially resulting in misclassification of equilibrium conditions as
degraded with RS vegetation indices.
Using a statistical approach to classifying the sites without the guidance of the RS
degradation labels, the plots were grouped into two distinct clusters. These were
distinguished based on stable soil properties which would be expected to be associated
with good soil health, such as total C, N, P and pH. The clustering has some overlap with
the equilibrium plots in both sites and therefore provides an indication of a reduced
number of soil properties that could be used to guide targeting efforts for restoration.
The cluster analysis revealed some consistent patterns within the soil data and some
agreement between the clustering and the classification derived from remote sensing
with seven out of nine and 12 out of 16 equilibrium sites attributed to the same cluster
at Kuresoi and Nyando, respectively.  These 'equilibrium' clusters were characterised by
higher soil P and C contents at both sites, suggesting that these clusters are more fertile.
pH was also an important variable in the two clusters at both sites, but with lower pHs
featuring in the Nyando site and higher pHs at Kuresoi. This reflects the different soils
present in the two areas: soils at Nyando are prone to salinisation and tend to have a
higher overall pH compared to the more acidic soils at Kuresoi, so it appears that what
we are seeing in the 'equilibrium' clusters is the inclusion of more favourable, slightly
acid pHs at both sites.  Of the transient variables, the enzymes PHO, GLC and XYL
featured in the 'equilibrium' clusters at both sites.  Both GLC and XYL are key for
breaking down cellulose and releasing energy for the soil microbial community, while
PHO plays an important role in releasing P from organic matter for plant uptake. Their
presence here could indicate that C and P are more limiting in the 'equilibrium' cluster.,
however, there is a lack of corroboration for this in the macronutrient data with C:N, C:P
and N:P ratios showing no significant difference (p <0.-5) between clusters at either site.
Our work points to the need to combine remote sensing techniques with field surveys,
but with a reduced set of measurements. Remote sensing is a powerful tool and provides



a cost-effective methodology for soil degradation assessment at ever increasing
resolution, but it is prone to error. This is particularly the case in landscapes with highly
heterogeneous smallholder grazing lands, such as those in Nyando and Kuresoi, where
vegetation cover and greenness, may be affected by intense grazing pressure resulting in
the misclassification of some sites, as shown in our work. On the other hand, field-work
is expensive and requires laboratory support, which is also costly and not always
available in SSA. However, our work demonstrates that utilising a relatively small set of
soil variables (soil microbial C, total C and Total N) can provide additional support for
classifications derived from remote sensing.

## Conclusion

Remote sensing was able to map grassland degradation over large areas of western
Kenya and offers the potential for cost-effective and dynamic monitoring. However, it
aligned only with a small subset of soil parameters, with soil microbial C being the only
parameter which consistently reflected changes in the degradation classes identified
from RS in both Nyando and Kuresoi. Additionally, some soil variables reflecting soil C
and N status did relate to the degradation classes at one or the other site. We expect that
variability in livestock grazing patterns and local climatic differences may have led to
some of the miss-classifications by RS.

The statistical clustering produced two clusters at each of the sites based on stable and
transient or dynamic soil properties. The clusters at each of the sites largely reflected
differences in nutrient status and biogeochemical cycling, particularly P and C contents
and PHO, GLC and XYL concentrations, with seven out of nine and 12 out of 16
equilibrium sites attributed to the same cluster at Kuresoi and Nyando, respectively.
Our research demonstrates the potential power of RS approaches to the assessment of
soil degradation, allowing temporally and spatial patterns of degradation to be resolved,
but also suggests that sampling a small additional set of soil variables that pertain to
biogeochemical cycling (soil microbial C, total C and Total N) can provide additional
support for the classification, identifying degraded soils and helping to target
restoration efforts..

## Author contributions

JQ and MR led the writing of the paper, MR led the study and secured the funding for the
work, along with RDB, JH and JQ. All authors contributed to the manuscript. In addition,
GY and GB carried out the remote sensing analysis and contributed to the data analysis,
MG carried out data analysis, KK, EF, YO, BN PDB, AO collected field data.

## Acknowledgements

We acknowledge funding from the UK Biotechnology and the Biological Sciences
Research Council (UKRI-BBSRC), through the Global Challenges Research Fund (GCRF)
under Agri-systems research to enhance rural livelihoods in developing countries [grant
number BB/S014934/1]. SML and KK acknowledge the CGIAR trust fund for funding
received through the Science Programmes Climate Action, Multifunctional Landscapes,
and Sustainable Farming.

## Dedication

We dedicate this paper to Joseph Hitimana, committed scientist and educator, and
critical to the success of this work who died before this paper could be published.



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
