# Peer review of "Soil degradation assessment across tropical grassland of Western Kenya"

_EGUsphere, 2025_

## Author Comment (AC1)

Many to the reviewer for the helpful comments

*General comment*

This manuscript tackles an important and timely question: whether multi-year satellite data on phenology (based on NDVI/EVI/NDWI processed with TIMESAT) can classify soil degradation states in smallholder grasslands and meaningfully relate to on-the-ground soil condition. The study spans two Kenyan landscapes and couples a remote-sensing classification (2013–2018) with field sampling at 90 sites (Oct–Nov 2019) for a suite of physical, chemical, and microbial variables (0–10 cm). The overall conclusion—that only microbial biomass C (and to a lesser extent bulk density) consistently aligns with the remote sensing classes—has practical implications for monitoring and restoration. However, several aspects of the methodology need clarification or strengthening before the evidence can fully support the claims.

*Major concerns.*

The paper mixes sensors (Landsat TM/OLI and Sentinel-2) and resamples to 10 m, but the harmonization/preprocessing steps are not fully described.

Response:

We thank the reviewer for their comments. We have now added the steps used to harmonise Landsat TM/OLI and Sentinel-2 imagery. Also, provided is a table that summarise the sensor by acquisition dates. New text below.

'To analyse the structural characteristics of grasslands supporting smallholder communities in Kuresoi and Lower Nyando, we implemented a time-series seasonal analysis that classified the stages of degradation on grasslands. We used 35 satellite image scenes from the archives of the European Space Agency (ESA 2016) and United States Geological Surveys (https://earthexplorer.usgs.gov/) (Table 1). The selection of different sensors was necessary to: i) fill missing dates from the Sentinel collection which had higher spatial resolution but shorter temporal resolution and ii) to maintain consistency in annual seasonal sampling between 2013 and 2018. The final satellite imagery was from Landsat-Thematic Mapper (TM) L2, Landsat Operational Land Imager (OLI) L2 and Sentinel-2 sensors L2A. Level 2 images are Analysis Ready Data (ARD) which are atmospherically corrected surface reflectance data and therefore free from the effects of haze and water vapour. Landsat-TM and OLI images were acquired with a spatial resolution of 30m, while Sentinel-2 images had a spatial resolution of 10m. The decision to select and process high-resolution imagery is due to the focus on smallholder dairy farms which are associated with grazing lands that are often less than 1ha and therefore easier to detect with higher resolution imagery. For LandSat-TM scenes, we downloaded blue (band 1), red (band 3), near-infrared (band 4), and shortwave infrared (band 6) spectral bands from USGS earth explorer repository. For Landsat-OLI scenes, we downloaded blue (band 2), red (band 4), near-infrared (band 5) and shortwave infrared (band 6). For the Sentinel-2 scenes, we downloaded blue (band 2), red (band 4), near infrared

(band 8) and shortwave infrared (band 11). We loaded the individual bands into RStudio using the raster package. All Landsat images were resampled to 10m with Sentinel-2 images as reference. We resampled the same spatial resolution because TIMESAT 3.3 requires all image scenes to have the same spatial resolution when creating raster stacks and before model fitting. No further image enhancements were applied because TIMESAT algorithm reduces negative biases arising from cloudiness by fitting the model to the upper envelope of the vegetation/water index data (REF). Despite these corrections, TIMESAT is unable to reduce negatively biased residuals related to surface anisotropy and sensor defects. However, we did not detect the effects of both during our analysis. Afterwards, we calculated NDVI values in each pixel by dividing the difference with the sum of near-infra red and red bands (Equation 1).To derive EVI values in each pixel, we applied correction factors and divided the difference between near-infrared and red bands with near-infrared band (Equation 2). We calculated NDWI in each pixel by dividing the difference with the sum of near-infrared and shortwave infrared (Equation 3).

**Table 1:** Summary of dates of acquisition of Landsat and Sentinel-2 imagery used for the determination of Normalized Difference Vegetation Index, Enhanced Vegetation Index and Normalized Difference Water Index.

| 2013 | 2014 | 2015 | 2016 | 2017 | 2018 |
|---|---|---|---|---|---|
| NA | 2014/01/25[1] | 2015/01/12[1] | 2016/01/08[2] | 2017/01/12[2] | 2018/01/22[2] |
| 2013/04/28[1] | 2014/04/01[1] | 2015/04/02[1] | 2016/04/27[2] | 2017/04/02[2] | 2018/04/10[1] |
| 2013/06/17[1] | 2014/06/18[1] | 2015/06/07[1] | 2016/06/06[2] | 2017/06/11[2] | 2018/06/11[2] |
| 2013/08/18[1] | 2014/08/21[1] | 2015/08/11[2] | 2016/08/25[2] | 2017/08/20[2] | 2018/08/05[2] |
| 2013/10/05[1] | 2014/10/25[1] | 2015/10/25[1] | 2016/10/29[1] | 2017/10/29[1] | 2018/10/03[1] |
| 2013/12/24[1] | 2014/12/11[1] | 2015/12/29[2] | 2016/12/23[2] | 2017/12/28[2] | 2018/12/18[2] |

[1] Landsat Thematic Mapper (TM) or Operational Land Imager (OLI) imagery
[2] Sentinel-2 imagery
*Note: Landsat images were resampled to 10m resolution.*

The specific land cover ESA product used for masking is not named or discussed in terms of accuracy/limitations for these mosaics.

To avoid sampling and analysing non-grassland areas, we masked urban, forest and water bodies. We used the ESA Climate Change Initiative (CCI) land cover vector layer (2015). We apply the masks during site selection for field visits and after the analysis of vegetation using time-series satellite images.

Terminology should be standardized (e.g., "Normalized Difference Vegetation Index," and clarify that your NDWI formulation uses NIR–SWIR, i.e., Gao-type, to avoid confusion with the original NDWI.)

All mentions of normalised differential vegetation index have been replaced with normalised difference vegetation index.

Degradation states are defined from average distributions of TIMESAT metrics and then selected by visual consistency with Google Earth, without an independent accuracy

assessment. At minimum, the manuscript should report a quantitative agreement/uncertainty analysis for the remote sensing maps.

We have inserted a section on remote sensing evaluation into the results (see below)

The evaluation of the ERU classification was implemented after field visitation in late 2019 and using 27 and 28  locations in Kuresoi and Lower Nyando respectively.  Farmers/land owners consent to access land parcels influenced the number of locations used in accuracy assessments. We compared field confirmations against the top-two classifying models of degradation states. One model was based on a combination of NDVI and NDWI (Table S2), and the second combined EVI, NDVI and NDWI (Table S3) for classification of degradation units. In Kuresoi, we also selected two models. One that combined EVI, NDVI and NDWI (Table S4) and one that used NDWI (Table S5) in landscape classification of degradation states.  The columns show the predictions from TIMESAT before farm visits and sampling and the rows represent the confirmations (truths) after farm visits in November 2019. The proportions of correct predictions for each class are also provided. Diagonals represent the number of correctly classified truths for each degradation unit.

**Table 2:** Accuracy assessment of classification of sampling locations determined from estimates of seasonal parameter values of normalised difference vegetation index, enhanced vegetation index and normalized difference water index for Kuresoi project sites. Columns are ground truth, rows are model prediction

|  | Productive | Degraded | Transitional | Correct prediction (%) | Predictions |
|---|---|---|---|---|---|
| **Productive** | **3** | 0 | 1 | 75 | **4** |
| **Degraded** | 3 | **5** | 8 | 31.3 | **16** |
| **Transitional** | 2 | 4 | **1** | 14.3 | **7** |
|  | **37.5** | **0** | **10** | **33.3** | 27 |
| **Count truth** | 8 | 9 | 10 |  |  |
|  |  | **Class accuracy** |  |  |  |
|  |  | **Overall accuracy** |  |  |  |

**Table 3:** Accuracy assessment of classification of sampling locations determined from estimates of seasonal parameter values of normalised difference water index for Kuresoi project site. Columns are ground truth, rows are model prediction

|  | Productive | Degraded | Transitional | Correct prediction (%) | Predictions |
|---|---|---|---|---|---|
| **Productive** | **3** | 0 | 1 | 75 | **4** |
| **Degraded** | 2 | **1** | 2 | 20 | **5** |
| **Transitional** | 3 | 8 | **7** | 38.9 | **18** |
|  | **37.5** | **0** | **10** | **40.7** | 27 |

| Count truth | 8 | 9 | 10 |
|---|---|---|---|

**Table 4:** Accuracy assessment of classification of sampling locations determined from estimates of seasonal parameter values of normalized difference vegetation index, enhanced vegetation index and normalized difference water index for Lower Nyando project site.Columns are ground truth, rows are model prediction

|  | Productive | Degraded | Transitional | Correct prediction (%) | Predictions |
|---|---|---|---|---|---|
| Productive | **1** | 0 | 2 | 33.3 | **3** |
| Degraded | 3 | **7** | 2 | 58.3 | **12** |
| Transitional | 6 | 3 | **4** | 30.7 | **13** |
|  | **10** | **0** | **25** | **42.9** | 28 |
| Count truth | 10 | 10 | 8 |  |  |

**Table 5:** Accuracy assessment of classification of sampling locations determined from estimates of seasonal parameter values of normalized difference vegetation index and normalized difference water index for Lower Nyando project site.  Columns are ground truth, rows are model prediction

|  | Productive | Degraded | Transitional | Correct prediction (%) | Predictions |
|---|---|---|---|---|---|
| Productive | **1** | 0 | 2 | 33.3 | **3** |
| Degraded | 1 | **6** | 1 | 75 | **8** |
| Transitional | 8 | 4 | **5** | 29.4 | **17** |
|  | **10** | **0** | **25** | **42.9** | 28 |
| Count truth | 10 | 10 | 8 |  |  |

With only ~35 scenes over 2013–2018 (≈ 6 per year) and no explicit treatment of cloud cover impacts on phenology fits, TIMESAT-derived timing metrics are likely uncertain.

Moreover, remote sensing labels summarize 2013–2018 whereas field sampling is in 2019 a gap that can be consequential in smallholder systems. These choices plausibly weaken soil–RS correspondence.  Several sections are overly detailed (lab methods) while key methodological choices (RS preprocessing, TIMESAT parameters) are terse.

We have now significantly expanded the description of the remote sensing work. See above. Given that we are publishing in a soil science journal we think that SOIL readers will be interested in the methods we used for soil analysis so do not propose to cut them.

The mismatch in timing of the remote sensing and the field programme are because the field programme had to be planned on the basis of the remote sensing, hence the last year we could use was 2018 for a 2019 field programme. We have added two sentences to the discussion to explain this.

'Additionally, the fact that the RS was used to plan the soil survey, meant that the RS images did not coincide with the survey dates. Given that we were using the RS data to consider seasonal shifts in vegetation indices over six years (Table S1) we do not think that an additional year of data would have changed our findings.'

Please also state whether a research permit/ethical clearance was obtained.

Yes, all research was ethically approved.

L78-80: Reporting a single stocking rate (1–2 cattle ha⁻¹) without nuance is misleading; please contextualize it by describing the different production systems.

We have updated this with more detail and supporting references in the revised manuscript.

'Smallholder systems in the highlands of Kenya have a range of stocking rates, typically expressed in Tropical Livestock Units (TLU) per hectare. For the for the Kenyan highlands between 1 and 1.4 TLU ha-1 are reported depending on the nature of the system (Bebe et al 2003), for Murang'a County to the south east of our study area 3-6 TLU ha-1 (Ortiz-Gonzalo et al, 2017) and for dairy cattle in Kiambu County to the west of our study area an average of 2.1 TLU ha-1 (Were et al 2025). '

L82-83; The discussion of soil degradation is overly simplified. Even if not the central objective, the manuscript should briefly address the complexity of degradation processes and site-specific drivers at the study locations

We wonder if the reviewer wrote this comment before reading the next paragraph which is devoted to degradation processes and describe the complexity of both the drivers and the process. We would argue that the introduction is not the place to describe all of the site specific drivers.

l172: Use 'difference' rather than 'differential' here."

Corrected

L310: This is an isolated citation.

Deleted